# Antimicrobial Peptides and Their Applications in Biomedical Sector

**DOI:** 10.3390/antibiotics10091094

**Published:** 2021-09-10

**Authors:** Afreen Sultana, Hongrong Luo, Seeram Ramakrishna

**Affiliations:** 1Center for Nanotechnology & Sustainability, Department of Mechanical Engineering, National University of Singapore, Singapore 117581, Singapore; affo.afreen123@gmail.com; 2Engineering Research Center in Biomaterials, Sichuan University, Chengdu 610064, China; hluo@scu.edu.cn

**Keywords:** antimicrobial peptides, biomedical application, functions of AMP, implementation techniques, antimicrobial activity, anti-inflammatory, anticancer, immunomodulatory effect, wound healing

## Abstract

In a report by WHO (2014), it was stated that antimicrobial resistance is an arising challenge that needs to be resolved. This resistance is a critical issue in terms of disease or infection treatment and is usually caused due to mutation, gene transfer, long-term usage or inadequate use of antimicrobials, survival of microbes after consumption of antimicrobials, and the presence of antimicrobials in agricultural feeds. One of the solutions to this problem is antimicrobial peptides (AMPs), which are ubiquitously present in the environment. These peptides are of concern due to their special mode of action against a wide spectrum of infections and health-related problems. The biomedical field has the highest need of AMPs as it possesses prominent desirable activity against HIV-1, skin cancer, breast cancer, in Behcet’s disease treatment, as well as in reducing the release of inflammatory cells such as TNFα, IL-8, and IL-1β, enhancing the production of anti-inflammatory cytokines such as IL-10 and GM-CSF, and in wound healing properties. This review has highlighted all the major functions and applications of AMPs in the biomedical field and concludes the future potential of AMPs.

## 1. Introduction

Microorganisms can be useful or hazardous, and some microbes show unrecognizable effects. Microbes with beneficial effects are important and are utilized for CO_2_ fixations, the degradation of complex organic compounds into basic molecules, fermentation, and many more applications. Furthermore, microorganisms can have a harmful effect, causing infections or diseases such as anthrax, conjunctivitis, ringworm, influenza, and many more [1]. Treatment of such health issues caused by microbes can be performed by appropriate diagnosis followed by adequate medication or therapy [2]. Antimicrobial compounds are considered a convenient solution for such health issues and can be of low molecular weight (LMW) or high molecule weight (HMW). Diverse relationships have been observed between the molecular weight and antimicrobial activity of these compounds. However, the majority of researchers report an increase in antimicrobial activity with the decrease in the degree of polymerization (i.e., molecular weight) [3]. It is expected that the adhesion of HMW antimicrobial polymers to a negatively charged bacterial cell membrane should be highly effective in comparison to the adhesion of LMW antimicrobial polymers. Instead, contradicting results are found, according to which LMW antimicrobial polymers have greater biocidial activity [4]. Usually for bacterial infections, antibiotics are the most commonly used drug for treatment, which show adverse effects such as reducing immunity, increasing susceptibility to infections, and developing resistance against antimicrobial agents [5].

Excess use of antibiotics results in antimicrobial resistance, which has devastating effects. This has led to the increase in demand for low molecular weight antimicrobial peptides, which are effective against antimicrobial-resistant bacteria [6].

Antimicrobial peptides (AMPs) are short-chain (5 to 100) amino acids that possess the ability to counter microbial attacks or any infective agent in all living organisms [7]. AMPs act as the first line of defense to disrupt bacterial, fungal, yeast, viral, and even cancer cells [8]. Most AMPs are found to be cationic, and are considered promising agents due to their distinctive mode of action and the inability of microbes to develop resistivity against them [9,10]. However, some of the AMPs are also anionic in nature due to the acidic polar residues, and these consist of short Asp-rich sequences [11]. More than 2500 AMPs are present in nature, which are broadly grouped into four categories: α helical, β sheet, extended, and αβ mixed antimicrobial peptides [12,13]. α helical AMPs possess a distance of about 0.15 nm between two adjacent bond angles (for example, LL-37) [14]. The β Sheet peptides are rigid structures that are more organized in an aqueous solution and do not show conformational change on membrane linkage (for example, Gomesin) [15]. Extended peptides comprise a high amount of arginine, tryptophan, proline, and/or histidine residues (for example, indolicidin and α1-purothionin) [16,17,18]. On the basis of activity, AMPs can also be classified as antimicrobial, antiprotozoal, anticancer, insecticidal, and antiparasitic [19]. AMPs can be harvested from invertebrates, vertebrates, and plants [20]. The first AMPs derived from certain sources were as follows: Cecropin from Hyalophora cecropia [21], defensis from rabbit [22], purothionin from wheat flour [23], and Gramicidin D from Bacillus brevis [24]. Among all the sources, some of the most valuable sources of AMPs with future prospects are marine animals [25], wild plants and weeds [26], and the black soldier fly [27]. A comparison between the antibiotics and AMPs is mentioned in Table 1.

Similar to AMPs, a group of polymers exists, known as synthetic antimicrobial oligomers, which are cationic and amphiphilic in nature [32]. These antimicrobial agents are fabricated to imitate AMPs [33]. However, these oligomers have a drawback of heterogeneity and innate toxicity [34]. Thus, antimicrobial peptides have the advantage of low toxicity over the synthetic antimicrobial oligomers and are highly effective in microbial disruption [35]. Recent studies have shown the impactful results of using AMPs in the biomedical sector. This article is different from various other papers on AMPs because it not only includes information about the factors affecting the functioning of AMPs and their important role, but along with it, focuses on the implementation techniques of AMPs in the biomedical sector. In this review, smart and intelligent delivery methods are also considered. Thus, this review could be helpful for readers that require compact information about the role of AMPs in biomedical applications. The schematic representation of this review is shown in Figure 1.

## 2. Factors Affecting the Functioning of AMPs

An upgraded database of the Collection of Antimicrobial Peptide comprises 18.7% helical structures of AMPs, 18.9% β-strand, and 60.1% mixed or coil [36]. The assortment of AMP structures suggests the diverse mechanism of action in the inhibition of microbial cells beyond pore formation. Characteristics such as the secondary structure, charge, hydrophobicity, hydrophobic moment, amphipathicity, polar angle, and peptide length are responsible for modulating the inhibitory mechanism of AMPs against microbes [37,38].

### 2.1. Secondary Structure

Secondary structure plays a significant role in improving the antimicrobial activity of AMPs. It is observed that among all the structures, the α helical structure promotes the insertion of antimicrobial peptides into the cellular membrane due to its facially amphiphilic structure [39]. In a previous investigation, two AMPs were compared, C5A (single helix) and AH (structure consisting of a hinge in between two short helixes), among which C5A had a more uniform helical structure. Analysis showed that a 10 nm concentration of C5A was capable of lysing lipid vesicles, whereas AH showed comparatively less potency [40].

### 2.2. Charge

Most of the AMPs are cationic in nature ranging between +2 and +9 [41]. In addition, Porto reported a range between +3 and +9 [42]. Charge plays a critical role in AMPs’ activity, which is reduced beyond the optimum level of charge [43]. Removal of the cationic terminus from melittin reduced the hemolytic activity and the ability to bind the membrane and can be used to modulate membrane selectivity [44]. Cationic AMPs bind with membranes consisting of negatively charged lipids due to an electrostatic interaction [45]. Anionic AMPs have a net positive charge but also consist of a few cationic residues, which interact with the membrane in a similar manner to cationic AMPs [46,47].

### 2.3. Hydrophobicity

Microbial membranes are protected from exogenous matter such as polysaccharides, proteins, and peptides due to its hydrophobic characteristics, but AMP has the ability to interact with the microbial membrane [48]. For instance, magainin has the ability to inhibit the growth of Gram-negative bacteria, but the analogs with improved hydrophobicity have shown effectiveness against Gram-positive bacteria as well [49].

### 2.4. Amphipathicity

Amphipathic peptides are ubiquitously present in nature with variation in their attributes [50,51]. The amphipathicity of AMPs can be defined as the ability to survive in solution under both hydrophobic and hydrophilic conditions [52]. Amphipathicity helps in improving inhibitory activity against microorganisms [16]. Narayana designed a trypsin-rich AMP (WW291) and studied the antimicrobial activity of eight permutated sequences (WW291 to WW298). From the results, he concluded that WW295 has increased antimicrobial activity four-fold against Klebsiella pneumonia. The appreciable activity of WW295 was supposed to be due to the presence of hydrophobic residues at the bottom and two hydrophilic amino acids at the top [53].

### 2.5. Hydrophobic Moment

The hydrophobic moment is similar to the electric dipole moment, and it can be defined as the lack of equality of hydrophobicity in the AMP’s structure [54]. The hydrophobic moment shows changes in rearranging the sequence [55]. With an increasing hydrophobic moment, antimicrobial peptides’ activity is found to be enhanced. Myxinidin was harvested from hagfish, then its analogues were developed, which are proved to serve as templates for the therapeutic agent. In this research, four AMPs were studied—myxinidin, myxinidin1, myxinidin2, and myxinidin3. Among all the antimicrobial peptides, myxinidin2 and myxinidin3 had the highest hydrophobic moments, which were 0.410 and 0.414, respectively. This factor has shown direct proportionality with the minimum inhibitory concentration (MIC) to kill microorganisms [56].

### 2.6. Polar Angle

The polar angle of AMPs can be defined as the relative proportion of polar to non-polar residues of the helix. The polar angle will be 180° if the number of polar residues is equal to non-polar residues. The polar angle is higher, if the number of hydrophobic residues is higher than hydrophilic residues, and visa-versa [57]. A lower polar angle signifies the better ability of the peptide to permeate the membrane [58]. This is also proved by another study, in which two staphylococcal peptides, warnericin RK and PSMα, were isolated and their anti-legionella activity was analyzed. Results suggested that the higher polar angle (around 140°) of both the peptides led to the need for aggregates of a bigger size to disrupt the microbial membrane [59].

### 2.7. Peptide Length

Peptide length is also among the critical attributes that affect the activity efficiency of AMPs. Juba investigated the effect of peptide length on membrane disruption using full-length cationic AMPs (NA- CATH) and truncated isomers (L- ATRA-1A and D- ATRA-1A). The activity of these peptides was tested against *Escherichia coli *and* Bacillus cereus*. It was observed that NA-CATH has higher antimicrobial potency than truncated peptides. It was also noted the peptide length affected the mechanism of action of peptides, and the full-length peptide disrupted the membrane via liposome lysing, whereas the truncated peptide caused liposomes’ exudation followed by fusion and aggregation [60].

## 3. Functions of AMPs

### 3.1. Disruption of Bacteria

AMPs have antimicrobial activity against a vast range of microorganisms and follow membrane disruption or non-membranolytic destruction of microorganisms [61,62]. In a previous study, stomoxynZH1 was extracted from Hermetia illucens via the RNA extraction method, which showed the inhibitory activity against *Staphylococcus aureus, E. coli, Rhi-zoctonia solani*, and *Sclero-tinia sclerotiorum* [63]. AMPs also play an important role in fighting against antibiotic-resistant bacteria. Polymyxin B has the capability of inhibiting the growth of multidrug-resistant *Pseudomonas aeruginosa, Klebsiella pneumoniae*, and *Acinetobacter baumannii* [64,65,66]. LS-sarcotoxin and LS-stomoxyn isolated from Lucilia sericata were found to inhibit 90% of *Salmonella enterica, E. coli, Acinetobacter baumannii, *and* Enterobacter cloacae* [67]. Some of the research proving the antimicrobial activity of AMPs are mentioned in Table 2.

### 3.2. Antifungal Activity

AMPs possess inhibitory activity against fungi [78]. In previous research, Roscetto isolated VLL-28 from an archaeal protein and examined the antimicrobial efficacy against 10 variants of *Candida* species. Among all the species, VLL-28 showed the lowest minimum inhibition concentration of 44.25 μg/mL against *C.tropicalis* 54 and *C.tropicalis* 2 and the highest MIC of 177 μg/mL against *C. glabrata* 28 and *C. glabrata* 34. Hence, it was concluded that VLL-28 has antifungal activity against planktonic cells and mature biofilms of *Candida* species [79]. AMPs with an antifungal property are mentioned in Table 3.

### 3.3. Antiviral Activity

The competency of the virus to replicate overwhelmingly is one of the causes of concern for life-threatening effects [92]. AMPs are effective and have minor side effects, which make them a potential alternative for the treatment of viral infection [93]. Zhang examined the antiviral potency of DP7 (VQWRIRVAVIRK) against *SARS coronavirus*. He claimed that DP7 has the ability to inhibit SARS-CoV and SARS-CoV-2 infection by examining its activity via cell receptor ACE2. He also reported that 104 μg/mL and 73.625 μg/mL (50% inhibitory concentration) of DP7 is required for inhibiting the SARS-CoV and SARS-CoV-2 pseudovirus [94]. Some of the research proving the antiviral activity of AMPs is mentioned in Table 4.

### 3.4. Inhibition of Cancer Cell Growth

Treatment of cancer has unbounded usage due to the easy development of resistance and the toxicity issue [104]. This has led the focus towards AMPs, which have the ability to resist cancer growth. Zhao reported the anticancer activity of the HPRP-A1 peptide isolated from Helicobacter pylori [105]. Further, the combined effect of iRGD (homing peptide) and HPRP-A1 were examined for enhancement in anticancer activity. Furthermore, the results suggested that iRGD helped in improving the penetration of HPRP-A1 on A549 MCS [106]. L-K6 is reported to be capable of killing MCF-7 breast cancer cells via nuclear disruption without cell surface disruption [107]. AMPs with the ability to inhibit cancer cell growth are mentioned in Table 5.

### 3.5. Immunomodulatory Effect

Several cationic AMPs are observed to have immunomodulatory characteristics due to two main reasons: Stimulation capability to induce chemokines and decrease in the release of undesirable pro-inflammatory cytokines [119,120,121,122]. In an investigation, clavanin A was modified to clavanin MO by inserting hydrophobic residues into the conserved oligopeptide FLPII to obtain both antimicrobial and immunomodulatory properties in the same peptide. The immunomodulatory effect of AMPs clavanin-MO isolated from marine tunicate was studied on the murine macrophage. It was observed from the results that clavanin-A and clavanin MO-treated samples showed an increase in the production of IL-10 (anti-inflammatory cytokine) and a decrease in IL-12 (a pro-inflammatory cytokine). Hence, it is concluded that clavanin-MO modulates the innate immune system due to its ability to stimulate leukocyte recruitment at the infection area and produce GM-CSF, IFN-γ, and MCP-1 [123]. A short synthetic peptide of 12 amino acids (RR) was designed with improved selectivity, antibacterial activity, and lesser toxicity. The immunomodulatory effect was analyzed by the results of acidity on antibacterial activity via measuring MIC, and the mechanism of bacterial inhibition was also explored in this study. Among all the analogues of RR, D-RR4 showed the lowest MIC_50_ and MIC_90_, which were 2 and 4 µM against *P. aeruginosa *and* A. baumannii* [124]. Some of the studies proving the immunomodulatory activity of AMPs are mentioned in Table 6.

### 3.6. Anti-Inflammatory Effect

AMPs are good candidates for suppressing inflammatory activity. The anti-inflammatory property of melectin was harvested from Melecta albifrons (bee venom), and cationic AMPs was studied via qRT-PCR. In this assessment, four groups were considered: Control (untreated human fibroblast), melectin-treated fiberoblast, *S. aureus*-infected fibroblast, and fibroblast treated with melecitin in the presence of *S. aureus* infection. In the melictin-treated group, no effect was observed in the reduction of pro-inflammatory cytokines. Meanwhile, the *S. aureus*-infected human fibroblast showed the maximum reduction in the release of TNFα, IL-8, IL-6, and IL-1β in melectin (5 µM) treated samples [134]. AMPs possessing anti- inflammatory activity are mentioned in Table 7.

### 3.7. Wound Healing

Several reports have suggested a wound healing property of AMPs [141]. Wound healing activity of recombinant P-LL37, which is derived from the human cathelicidin antimicrobial peptide LL37, was studied on dexamethasone-treated mice. Mice were cleaned and a 5 mm-diameter wound was created on the dorsal surface under aseptic conditions. Two times a day, wounds were treated with LL37 and sterile water. A re-epithelialization study suggests that the keranocytes layer is complete in AMPs-treated samples, whereas in untreated samples, the keranocytes layer is incomplete. A higher number of blood vessels was observed in treated samples compared to untreated samples [142]. Several research studies on the ability to effectively heal wounds are mentioned in Table 8.

## 4. Implementation Techniques

### 4.1. Impregnation of AMPs

The incorporation of AMPs into multilayers is the most commonly studied technique because of its ability to reduce the loss of antimicrobial activity and the fact that it requires a low quantity of AMPs [150,151]. The layer-by-layer technique was used to incorporate AMP (nisin Z) into polyelectrolyte multilayers (PEM) made up of chitosan and carrageenan (CAR). This study involved three groups: PEM without AMP, PEM with AMP, and PEM with an AMP/CAR outer layer. Among all the groups, PEM consisting of nisin showed the highest antimicrobial activity against S. aureus and MRSA. During adsorption, some of the nisin Z was lost, and 0.89 ± 0.064 µg cm^−2^ was retained [152]. Antimicrobial peptides-embedded cotton gauze (18 mm × 18 mm), functionalized with chitosan and alginic acid sodium salt, was examined as a novel method of antimicrobial wound-dressing. This method involved the soaking of cotton gauze in chitosan and alginic acid sodium salt for a 5-min duration. It was then immersed in AMP, followed by washing with deonized water and coating with chitosan. Four antimicrobial peptides were studied: hBD-1, β-Defensin-1, human dermaseptin, Cys-LC-LL-37, and Magainin 1. The lowest MIC was observed for Magainin 1 against S. aureus and K. pneumoniae. The release study recorded the release after 6 h of immersion and also noted that Dermaseptin had the fastest release and Magainin had the slowest among all the samples. It was found that after 24 h, only 75% of AMPS was utilized from the cotton gauzes [150]. Cubosome is a nanoparticle representing a lipid biomimteric environment that has been studied for the incorporation of AMPs [153]. Gramicidin A and alamethicin were incorporated into cubosome (composed of monoolein, monopalmitolein, monovaccenin, and 1-(7Ztetradecenoyl)-rac-glycerol). The range of AMP concentration was considered to be from 0 to 10% and it is observed that with the increase in concentration, cubic symmetry was decreased. The loss of structure can be prevented by using the lipid bilayer, which is approximately equal to the peptide length [154]. Methods of incorporating AMPs into a matrix are shown in Figure 2.

### 4.2. Scaffolding

Scaffolding is a three-dimensional porous solid biomaterial that acts as a matrix to hold bioactive compounds [155]. This method can be used to incorporate AMP to obtain the desired features. In an investigation, LL37, melittin, buforin, MM1, AR23, and RV23 were scaffolded in protofibril to study the immunomodulatory potency. All of the AMPs formed square columnar lattices. These scaffolds were incubated with human pDCs and the production of IFN-α was recorded. The results suggest that LL37-dsDNA and buforin-dsDNA stimulate significantly higher production of IFN-α compared to the control. Scaffolding of AMPs in ordered nanocrystalline complexes helps in amplifying the production of cytokines to modulate immunity [156]. Ye prepared a scaffold from intrafibrillar mineralized collagen via the biomimetic technique incorporated with GL13K peptides to treat bone defects. This research involves the study of collagen mineralized for four different durations (1, 2, 4, and 8 days). The morphology suggested that 4 days and 8 days of mineralization led to homogeneity and a high degree of mineralization. Mineralization was more useful in the proper loading of AMP in comparison to non-mineralized scaffolds. The release of GL13K was not observed for up to 14 days, but a drastic release was reported from 21 to 28 days, which may be due to collagen hydrolysis. The hydrophobicity of the scaffold was improved by coating it with GL13K, which suggests stability in the presence of water. GL13K scaffolds, which were mineralized for two days or more, reduced *S. gordonii* viability in a better way compared to others [157]. Rabanal and a few others developed a scaffold from lipopeptide, which is found to possess improved antimicrobial activity against *P. aeruginosa*, *E. coli*, *S. aureus*, and *E. faecalis* [158].

### 4.3. Electrospinning

Electrospinning is a novel technology that consists of a high-voltage power supply, a syringe, which is the producer of nanofiber from the polymeric solution (includes pump and needle), and a collector surface on which the polymeric solution is coated during the process [159]. Electrospinning is an effective and inexpensive technique used in biological and medical applications, which involves the incorporation of a strong electric field to develop nanofibers [160]. Electrospinning technology is used for biomedical purposes such as electrospun prosthetic heart valves [161]. Shortcomings in such innovations concern microbial deterioration, which can be avoided with the use of antimicrobial agents. Various studies have proved the use of AMPs instead of antimicrobial agents to prevent the risk of infection of electrospun materials [162]. For instance, Cm-p1 is derived from the Caribbean Sea mollusk, which was added in PVA and used in electrospinning a solubilized solution of Cm-p1 with 2.5%, 5%, and 10%, *w/v* concentrations in 0.5 mL of deionized water. Further, a polymeric solution was prepared by adding 50 g of PVA [163]. This solution is pumped at a flow rate of 0.2 mL.h^−1^ via a syringe at 15 kV of voltage with a distance of 10 cm between the needle tip and collector. The releasing property of the nanofiber was studied using RP-HPLC and the sustainable release of Cm-p1 was observed for up to 48 h. Only Cm-p1 10%-PVA inhibited *C. albinas* growth, resulting in a 3.94 mm diameter of the zone. The immune response was studied via the release of cytokines IL-6 and TNF-α by RAW 264.7 macrophages cells. These cytokines had the ability of low induction by Cm-p1 10%-PVA [164]. Similar electrospinning conditions were considered in another study. Pleurocidin in four different concentrations (0.03 wt.%, 0.06 wt.%, 0.12 wt.%, and 0.25 wt.%) was mixed with 12% PVA. PVA with 0.25% pleurocidin inhibited *E. coli* growth within 5 h of storage. The antimicrobial property in apple cider was also assessed, where results suggest that within 14 days, PVA with 0.25% pleurocidin inhibited *E. coli* growth to an undetectable level. They concluded that electrospinning is useful in controlling the release of antimicrobial peptides [165]. Figure 3 represents the schematic diagram for the electrospinning process.

## 5. Applications in Biomedical Sector

### 5.1. Vertebrate-Derived Antimicrobial Peptides

Vertebrates are mammals with spinal cords, such as humans, marine animals, birds, and other animals. AMPs isolated from vertebrates are studied due to their potency against a wide range of microorganism. The most commonly found and highly investigated antimicrobial peptides are defensins and cathelicidins [166]. Defensins are positively charged antimicrobial peptides composed of 29–34 amino acids in a β-sheet structure [167]. They are categorized into three groups—α-, β, and θ-defensins—and possess antimicrobial activity in the range of 0.5–5 μm [166,168]. A-Defensins are found in inflamed tissues [169], β-defensins are present in bovine neutrophils [170], and θ-defensins can be harvested from neutrophils and bone marrow [171]. Cathelicidin is a multifunctional peptide that has conserved pro-peptide sequences and is identified as an N-terminal signal peptide [172,173]. They are harvested from porcine intestines and bovine neutrophils [174,175]. These peptides have shown wound healing activity, immunomodulatory effect and apoptosis [176].

Shaat investigated serum α-defensins 1–3 and salivary α-defensins 1–3 for Behcet’s disease treatment. The receiver operating characteristic was noted as 0.743 and 0.936 for serum and salivary α-defensins, respectively, suggesting potential in the pathogenesis of Behcet’s disease [177]. Human defensins-5 has shown beneficial effects in treating dysbiosis [178]. Tracheal AMP (β-defensins) has shown resistance to Mycoplasma bovis, which is a respiratory pathogen [179]. Crohn’s disease can be eliminated by inducing β- defensins [180]. In patients suffering from periodontitis, β-defensins play a critical role in bone repairment [181]. Cole studied a peptide named retrocyclin (θ-defensins), which prevents infection by T-and M-tropic strains of HIV-1 [182]. θ-defensins has the capability to suppress inflammatory cytokines and is also used in therapy for the *herpes simplex virus* [183,184]. Retrocylin-2 has potency in treating *Influenza A Virus* [185].

The potency of cathelicidins LL-37 was examined on human MDMs and THP-1 cells, which were infected with Mycobacterium tuberculosis. These cells were infected with the M. tuberculosis H37Rv for four hours, followed by treatment with 1 µg/m of LL-37 for 24 h. It was concluded that LL-37 mediates the activation of autophagy via the P2RX7 receptor and inhibits the growth of Mycobacterium tuberculosis [186]. In another study, the Cathelicidins-inspired peptides have shown antifungal activity against *Fusarium, Aspergillus, Cryptococcus, Malassezia, Candida, *and* Talaromyces* [187]. Applications of vertebrate-derived AMPs are not limited to the few above-discussed AMPs; instead, there are many AMPs with great potential to treat infections and health issues. For instance, Scapularisin-3 and Scapularisin-6 isolated from Ixodes scapularis are reported to have a strong inhibitory property against *Fusarium culmorum* and *F. graminearum* [188].

### 5.2. Insect Derived AMPs

Among all sources of AMPs, insects are resistant to a wide range of microorganisms due to their tolerance to harsh living conditions [189]. Insect AMPs are found to have great potential in treating skin cancer [190]. CopA3 isolated from Copris tripartitus has proved the ability to treat gastric cancer and leukemia as well [117,191]. An evaluation of antibacterial and inflammatory properties of DLP2 and DLP4 isolated from Hermetia illucens and expressed in *P. pastoris* was performed. The time-dependent relation was recorded for antimicrobial activity against MRSA. It was observed that Log10 (CFU/mL) of *S. aureus* was reduced to 1.68‒1.89 and 1.06‒1.34 for DLP2 and DLP4, respectively. DLP4 destroyed 99% of *MRSA* within 2 h, while DLP2 took 6 h. It was also concluded that DLP4 has two- to four-fold higher activity than DLP2 against MRSA. In experimentation on mice, it was observed that DLP4 (7.5 mg/kg) has better efficiency against *S. aureus* inhibition than DLP2 (7.5 mg/kg). A smaller antibiotic effect was reported in treatment with DLP2 compareed to DLP4 [128].

### 5.3. Plant Derived AMPs

The smallest AMPs were derived from jatropha curcas with seven amino acids [192]. Commonly found AMPs in plants are the cyclotide family, thionins, the α-Hairpinin family, Hevein-like peptides, lipid transfer protein, knottin-type peptides, snakins, and plant defensins [193]. Thionin is a useful AMP for the development of a glucose detection biosensor. Salimi and coworkers induced thionin in multiwall carbon nanotubes to develop a sensor that selectively detects glucose depending on the cathodic peak current [194]. Another probe was fabricated as a lung cancer biomarker using the nanohybrid of graphene oxide-thionin-hemin-Au with a low detection limit of 0.026 pg mL^−1^. In this probe, graphene oxide acted as a supporting material in which thionin and hemin were immobilized followed by a reduction of silver particles by thionin [195]. Hu and coworkers harvested three novel cyclotides from roots and leaves of Hedyotis diffusa. Experimental data suggest the inhibition of invasion of LNCap cells, which confirms the anti-cancer effect [196]. An analogue of Cp-thionin II (KT43C) was isolated from cowpea and the study showed that the growth of *F. culmorum, P. expansum, *and* A. niger* is inhibited [197]. Neutrophil elastase-associated diseases can be prevented by using roseltide rT1, which is derived from *Hibiscus sabdariffa* [198]. Plant defensins MtDef5 extracted from Medicago truncatula have shown the ability to permeate the membrane of *F. graminearum* and *Neurospora crassa* [199]. Hence, AMPs harvested from plant sources have a wide spectrum of applications in the biomedical field with future prospects.

### 5.4. Microorganism Derived AMPs

AMPs derived from bacteria are usually termed bacteriocin, such as nisin [200]. Nisin is isolated from Gram-positive bacteria [201,202]. The anticancer effect of nisin has been reported [203,204]. Begde reported the immunomodulatory effect of nisin due to its ability to activate neutrophils [205]. Other AMPs from microorganisms such as enterocins DD28 and DD93, derived from Enterococcus faecalis, have shown inhibitory activity against MRSA [206]. Table 9 represents the biomedical applications of nisin.

## 6. Smart and Intelligent Delivery of AMPs

AMPs, though capable enough to prevent various biological and medical issues, still have limited commercial applications. The release of antimicrobial molecules in the presence or absence of infections (i.e., uncontrolled release) is one of the major concerns [210]. Novel drug delivery systems include smart and intelligent mechanisms that are capable of adjusting the release rate of drugs according to the physiological conditions of the patients [211]. Novel delivery systems are classified into four groups: Systems in which the release rate depends on a specific targeting moiety, activation-modulated delivery systems, systems in which the release rate depends on triggering agents, and systems with the programmed release of the drug [212].

Other than the uncontrolled release rate, factors such as low biocompatibility, low metabolic stability, low solubility, high susceptibility to degradation, and toxicity act as a hurdle in applications of AMPs (Figure 4) [213].

The incorporation of antimicrobial peptides into a nanostructure can work as a solution to these issues. Some of the nanosystems researched are metal nanoparticles [214], carbon nanotubes [215], liposomes [216], liquid crystalline particles [217], dendritic systems [218], aptamers [219], hydrogels [220], polymers [221], and cyclodextrins [222].

Among all, to combat barriers in the delivery of AMPs, nanofibers are considered an excellent method that can be produced by electrospinning or self-assembly [223]. Electrospinning is a process that involves the deposition of a polymeric solution on the substrate surface via an electric field [224]. Various applications of electrospinning involved in AMPs’ incorporation are mentioned in Section 4.3. The self-assembled system has a precise structure that involves nanocarriers for target delivery [225]. AMPs L-5 was incorporated into poly (ethylene glycol)-co-acrylic acid microgel, and the results suggest that this self-assembly significantly limited the release of AMPs, improving the anti-adherent property and antimicrobial activity [226]. These novel techniques have the potential to overcome challenges in applications of AMPs in the biomedical field.

## 7. Conclusions

AMPs act as bioactive compounds with huge potential in the field of the biomedical sector. AMPs are harvested from various sources, consisting of varying structures and characteristics. These AMPs are studied in depth to obtain a better understanding of the structure, mechanism of action, and applications. Various studies have proved the significance of AMP over other drugs for the treatment of infections and health issues. It has been claimed to be a better option for treatment due to its eminent anti-inflammatory effect, immunomodulatory activity, antimicrobial activity, anticancer activity, and wound-healing properties. On the contrary, AMPs have several disadvantages that limit the application of AMPs in the biomedical field, such as the uncontrolled release rate, which results in loss of the antimicrobial peptide and reduces its efficiency. The isolation of antimicrobial peptides is another challenge, which is a tedious and costly process. Preservation of activity of AMPs throughout its expected use is also a point of concern. AMPs are further under investigation to overcome these challenges by designing a better system with a controlled release rate, efficient functioning at desired conditions, and minimum or no sensitivity, to deliver expected results. The incorporation of AMPs into a matrix can be performed in several ways by altering the preparation conditions, which affects the end result and activity of AMPs. This article has discussed most of the recent studies on the properties and impregnation methods of AMPs, along with their scope in the biomedical sector.

## Figures and Tables

**Figure 1 antibiotics-10-01094-f001:**
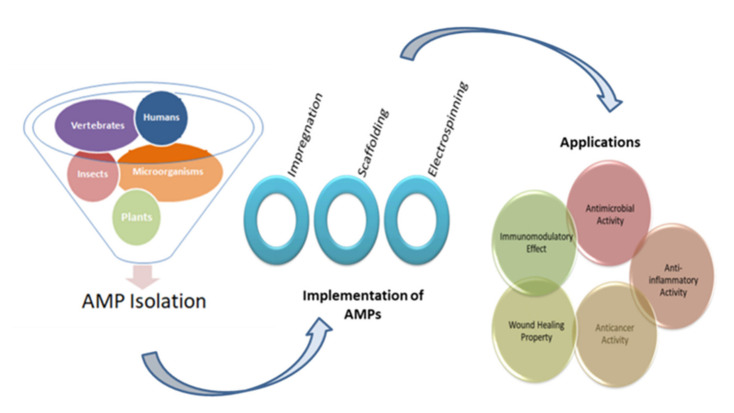
Diagrammatic representation of the five main sources for the isolation of AMPs, three widely used implementation techniques, and their applications.

**Figure 2 antibiotics-10-01094-f002:**
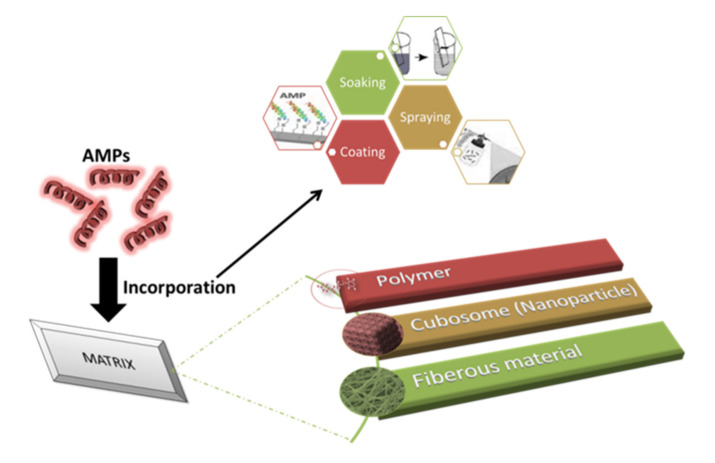
Impregnation of AMPs into compatible matrices. Incorporation of AMPs in a matrix is usually performed via soaking, coating, or spraying. The three common matrices are polymer, cubosome, and fibrous material.

**Figure 3 antibiotics-10-01094-f003:**
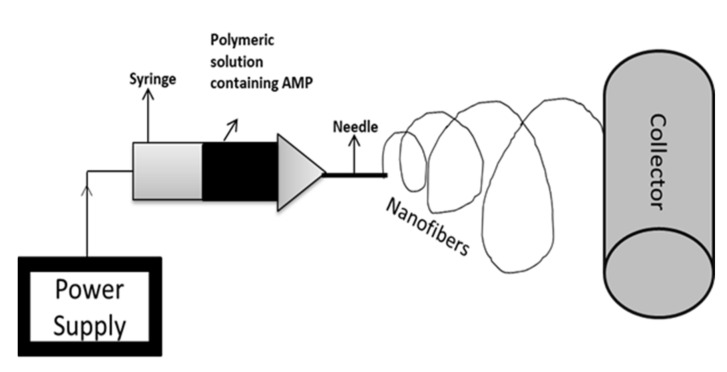
Schematic representation of electrospinning technique. Electrospinning is a nanofabrication method that involves the controlled flow of a polymeric solution placed in a syringe via a needle. The collector is the material on which nanofibers are coated.

**Figure 4 antibiotics-10-01094-f004:**
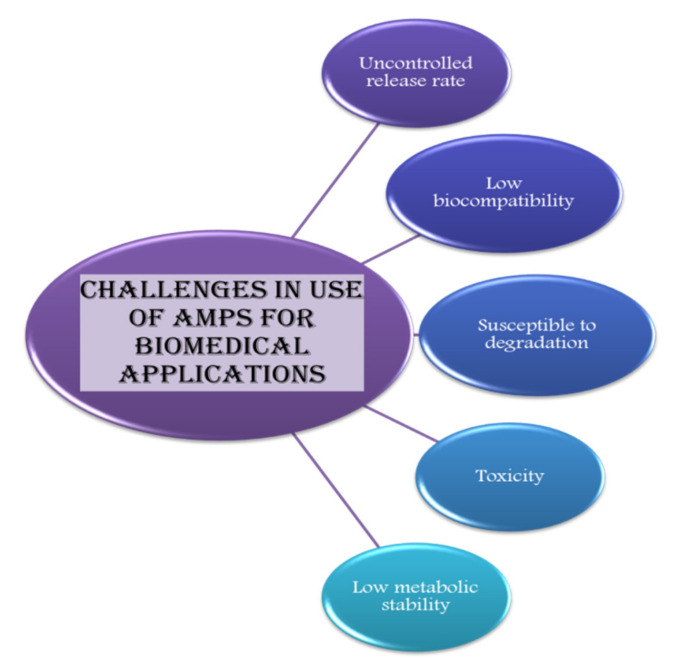
Amps can be used for various applications in the biomedical sector but its use is limited due to certain challenges, which are mentioned in this diagrammatic illustration.

**Table 1 antibiotics-10-01094-t001:** Comparative study between antibiotics and AMPs.

Characteristics	Antibiotics	AMPs	References
Similarities	Destruction of microorganism		[28,29]
Differences	No immunomodulatory effect observed	Has immunomodulatory effect	[30]
	Microorganisms easily develop resistivity against antibiotics	No easy development of resistivity	
	No inflammatory response	Has effective anti-inflammatory activity	[14]
	Effective at low concentration	High concentration is required	[31]

**Table 2 antibiotics-10-01094-t002:** List of some of the AMPs with putative antibacterial activity.

AMPs	Source	Bacteria Inhibited	Antimicrobial Activity	References
ZmD32	Corn	*E. coli, Bacillus subtilis, P. aeruginosa*, and *S. aureus*	50% Inhibitory concentration of ZmD32 ranged between 0.4 and 1.7 μM	[68]
LL-37	Human cathelicidin hCAP18	*Methicillin-resistant Staphylococcus aureus, methicillin-susceptible S. aureus, Vancomycin Intermediate Staphylococcus aureus (VISA) *and* Vancomycin Resistant Staphylococcus aureus (VRSA)*	Minimum inhibitory concentration was recorded as 64,128,64 and 256 µg/mL for *VISA, MSSA, VRSA*, and *MRSA*, respectively	[69]
Melimine and Mel4		*P. aeruginosa*	It took 4 and 30 min for Mel4 and melimine, respectively, to permeate through the cytoplasmic layer	[70]
Cecropin A	Moth	*Uropathogenic E. coli (UPEC)*	Incorporation of 0.25 µM^−1^ µM CecA with nalidixic acid was able to permeate through UPEC cell membrane by 15%.	[71]
BING	Japanese medaka plasma	*Broad spectrum including E. coli, Enterococcus faecalis, S. aureus and P. aeruginosa A*	Minimum inhibitory concentration of BONG ranged between 4 and 50 µg/mL	[72]
D-Cateslytin	Human	*Methicillin-susceptible Staphylococcus aureus, Methicillin-resistant Staphylococcus aureus, Pseudomonas micra, Pseudomonas intermedia *and* F. nucleatum*	Minimum inhibitory concentration of D-Cateslyt ranged between 8 and 24 μg/mL	[73]
Guavanin 2	guava	*E. coli, Listeria ivanovii *and* Candida parapsilosis*	Minimum inhibitory concentration for *E. coli, Listeria ivanovii*, and *C. parapsilosis* was recorded as 6.25, 50, and 50 µM	[74]
Thanatin		*E. coli*and*K. pneumoniae*	Thanatin replaces divalent cations from bacterial membrane and causes disruption	[75]
Temporin B	Frog skin	*Staphylococcus epidermidis*	Chitosan nanoparticles containing Temporin B showed 4 log reduction of *S. epidermis* compared to chitosan nanoparticles	[76]
Oncocin	Milkweed bug	*P. aeruginosa, E. coli and Acinetobacter baumannii*	Minimum inhibitory concentration of Oncocin was recorded as 0.125 to 8 μg/mL	[77]

**Table 3 antibiotics-10-01094-t003:** List of some of the AMPs with a putative antifungal activity.

AMPs	Source	Fungi Inhibited	Results	References
Tk-AMP-X1 and Tk-AMP-X2	Triticum kiharae	*Fusarium graminearum, Diplodia maydi *and* Fusarium verticillioides*	50% inhibition concentration of Tk-AMP-X1 and Tk-AMP-X2 range between 7.5 and 30 µg mL^−1^	[80]
OsAFP1	Rice	*Candida Albicans*	Inhibited *C. albicans* growth at 4µM concentration	[81]
LL-37	Human	*Aspergillus fumigatus*	After 30 min of incubation, LL-37 binds to mycelia and damages the cell wall	[82]
oAP2 and NDBP-5.7	Tityus obscurus and Opisthacanthus cayaporum scorpions	*C. Albicans*	MIC for Oap2 and NDBP-5.7 was recorded as 25 µM and 100 µM, respectively	[83]
NCR044		*Botrytis cinerea, Fusarium oxysporum, F. graminearum *and* Fusarium virguliforme*	50% of inhibition concentration ranged between 0.52 and 1.93 µM	[84]
ASP2397	Malaysian leaf litter	*A. fumigatus*	MIC was recorded as 0.78 µgmL^−1^	[85]
NoPv1	Synthetic	*Plasmopara viticola*	200µM of NoPv1 showed the complete destruction of *Plasmopara viticola*	[86]
Metchnikowin	Drosophila melanogaster	*F. graminearum*	50% inhibitory concentration of Metchnikowin was found as 1 µM	[87]
LBM 18	Pediococcus pentosaceus	*A. niger and Aspergillus flavus*	Within 2 days of incubation BLIS was able to cause destruction to *A. niger and Aspergillus flavus*	[88]
Penetratin	Synthetic	*C. albicans *and* C. glabrata*	50% inhibitory concentration was recorded in a range of 1 to 50 µM	[89]
APP	ppTG20	*Saccharomyces cerevisiae, C. albicans, A. niger, Trichopyton rubrum, A. flavus *and* Cryptococcus neoformans*	MIC was recorded as 8, 16, and 32 µM for *C. albicans *and* A. flavus, Saccharomyces cerevisiae*, and *Cryptococcus neoformans *and* A. niger *and* Trichopyton rubrum, respectively*	[90]
polybia-CP	Polybia paulista	*Candida strains*	MIC of antimicrobial peptide was recorded in a range between 4 and 64 µM	[91]

**Table 4 antibiotics-10-01094-t004:** List of some of the AMPs with putative antiviral activity.

AMPs	Source	Virus Inhibited	References
Melittin	*Apis mellifera*	Suppress the activation of cathepsin S	[95]
Lactoferrin	Mucosal secretions	SARS-CoV	[96]
HD-5	Human	Human papillomavirus	[97]
PD3, PD4, and RW3	Thrombin-induced human platelet and synthetic repeats of arginine-tryptophan	Vaccinia virus	[98]
Human β-defensin 3	Human	human immunodeficiency virus and herpes simplex virus	[99]
ALFPm3	*Penaeus monodon*	white spot syndrome virus	[100]
HS-1	*Hypsiboas semilineatus*	Dengue virus	[101]
Myticin C	Mussel	Herpes viruses	[102]
P9 (β-defensin-4)	Mouse	Influenza A virus H1N1, H3N2, H5N1, H7N7, H7N9, SARS-CoV and MERS-CoV.	[103]

**Table 5 antibiotics-10-01094-t005:** List of some of the AMPs that can be used in cancer therapeutic studies.

AMPs	Source	Significance	References
Poca A, Poca B and CyO4	*Pombalia calceolaria*	Reduced the breast cancer cell up to 80%	[108]
Aurein 1.2	Frog *Litoria aurea*	Among 54 cancer cells, 52 are inhibited in NCI testing method	[109]
Bmattacin2	*Bombyx mori*	Disrupted A375 and HCT116 cancer cells	[110]
Laterosporulin10	*Brevibacillus sp.*	MCF-7, H1299, HEK293T, HT1080, and HeLa cancer cells were disrupted	[111]
Dermaseptin-PD-1 and dermaseptin-PD-2	Phyllomedusine leaf frogs	Growth of H157, PC-3, and U251 MG cancer cell was inhibited	[112]
Scolopendrasin VII	Centipede	Reduction in viability of leukemia cells	[113]
Myristoyl-CM4	Synthetic	Activates caspase 9, caspase 3, and cleavage of PARP in breast cancer cells	[114]
K4R2-Nal2-S1		Binds with lung cancer cells and results in apoptosis	[115]
VLL-28	*Sulfolobus islandicus*	Inhibits murine and human tumor cells	[116]
CopA3	*Copris tripartitus*	Reduction in cell viability of gastric cancer cells	[117]
Pardaxin	*Pardachirus marmoratus*	Improved the activation of caspase-3	[118]

**Table 6 antibiotics-10-01094-t006:** List of some of the antimicrobial peptides suggesting an immunomodulatory effect.

AMPs	Source	Mechanism	References
Nisin Z	Gram positive bacteria	suppress LPS-induced pro-inflammatory cytokines	[125]
LL-37	Human	Reduces pro-inflamatory mediators	[126]
PMAP-23	Porcine	Induces production of IL-8 in porcine epithelial cells	[127]
Defensins-DLP2 and DLP4	*Hermetia illucens*	Decreases the pro inflammatory cytokines production	[128]
Epinecidin-1	*Epinephelus coioides*	Increased the expression of TNF-1	[129]
cNK-2	Chicken	Induces the expression of CCL4, CCL5 and interleukin(IL)-1β	[130]
Tilapia hepcidin (TH)2-3	*Pichia pastoris*	Produces certain short-chain fatty acids to improve immunity	[131]
CRAMP	Human	Increases TLR9 expression, which suppresses cardiac hypertrophy	[132]
cLF36	Camel lactoferrin	Reduces IL-2 and MUC2 expression	[133]

**Table 7 antibiotics-10-01094-t007:** List of some of the AMPs suggesting an anti-inflammatory effect.

AMPs	Source	Mechanism	References
Defensins-DLP2 and DLP4	*Hermetia illucens*	Induces the production of anti- inflammatory cytokines IL-10 and GM-CSF	[128]
cecropin-TY1	*Tabanus yao*	Inhibits the production of pro-inflammatory cytokines	[135]
SET-M33D	Synthetic	Reduces the production of TNF-α, COX-2 IL6, KC, IP10, MIP-1, iNOS, NF-κB	[136]
Papiliocin (Pap12-6)	Swallowtail butterfly	Decrease in secretion of NO, TNF-α, and IL-6	[137]
L-37	Humans	Reduces intestinal inflammation	[138]
Lipocalin 2 (Lcn2)	Epithelial and myeloid cells	Increases cytokine expression and NFκB activation	[139]
Hc-cath	*Hydrophis cyanocyntus*	Decrease the release of pro-inflammatory cytokine and neutrophil chemoattractant	[140]

**Table 8 antibiotics-10-01094-t008:** List of some of the AMPs that suggest wound-healing properties.

AMPs	Source	Mechanism	References
Cys-KR12	Human	Suppression the LPS-induced TNF-α	[143]
LLKKK18 (Analog of LL37)	Humans	Rapid wound closure and reduction in oxidative stress	[141]
Os	Synthetic	AMP encourages collagen deposition	[144]
1a(1-21)NH2	Frog skin	Activates epidermal growth factor receptor and STAT3 protein. Promotes migration of keratinocytes (HaCaT cells)	[145]
Defensin-1	Royal jelly	Promotes matrix metalloproteinase-9 secretion and increases migration of keratinocyte	[146]
DRGN-1	Komodo dragon	Stimulates the migration of HEKa keratinocyte cells and activates EGFR-STAT1/3 pathway	[147]
Tiger-17	Designed antimicrobial peptides	Promotes the release of TGF-β1 and IL-6 which aids in formation of tissue	[148]
Brevinin-2Ta	*Pelophylax kl. esculentus*	Angiogenesis process was observed	[149]

**Table 9 antibiotics-10-01094-t009:** Biomedical applications of nisin (AMPs derived from bacteria).

AMPs	Therapeutic Applications	Mechanism	References
Nisin A	Head and neck squamous cell carcinoma	Induces apoptosis which abbreviates tumor formation	[207]
Nisin A	Prevents skin cancer	Retards the DMBA induced skin carcinogenesis	[204]
Nisin A S29A, S29D and S29E	Prevents food borne infections	Inhibits the growth of *E. coli, Cronobacter sakazakii, Salmonella enterica*and*Typhimurium*	[208]
Nisin Z	Treatment of oral issues	Inhibits the growth of oral pathogenic bacteria	[209]

## Data Availability

Not applicable.

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
