# Peer review of "Antimicrobial Peptides and Their Applications in Biomedical Sector"

_antibiotics, 2021, doi:10.3390/antibiotics10091094_

Round 1
Reviewer 1 Report
Recommendation:
Comments: Publish after minor revisions.
This manuscript described antimicrobial peptides (AMP) in great details. The author firstly described AMP structure and property relationship. Then listed and discussed all functions of AMP. Finally, the author discussed about the application of AMP. This manuscript is well written and can be considered for publication after the authors address the following issues.
- Could authors compare AMP with synthetic antimicrobial oligomers and discuss in detail? It will raise more interest to audience.
- In terms of molecular weight, AMP is kind of low MW molecules. Could authors also discuss about high MW antimicrobial agents such as antimicrobial polymers? Comparing low and high MW antimicrobial agent will give audience a high-level understanding.
- In the last part of the manuscript, the author described a new method which might help to overcome AMP delivery issues. Is this the only effective method? Could authors described more about it?
Author Response
Thank you for your valuable suggestions. Considering them I revised my manuscript.
1. Could authors compare AMP with synthetic antimicrobial oligomers and discuss in detail? It will raise more interest to audience.
REPLY: Similar to antimicrobial peptides, a group of polymer exists known as synthetic an-timicrobial oligomers which are cationic and amphiphilic in nature [33]. These antimicrobial agents are fabricated to imitate antimicrobial peptides [34]. But these oligomers have a drawback of heterogeneity and innate toxicity [35]. Thus, antimicrobial peptides have an advantage over the synthetic antimicrobial oligomers of low toxicity and are highly effective in microbial disruption [36].
2. In terms of molecular weight, AMP is kind of low MW molecules. Could authors also discuss about high MW antimicrobial agents such as antimicrobial polymers? Comparing low and high MW antimicrobial agent will give audience a high-level understanding.
REPLY: Antimicrobial compounds are considered as convenient solution for such health issues which can be of low molecular weight (LMW) or high molecule weight (HMW). Diverse relation has been observed between molecular weight and antimicrobial activity of these compounds. But the majority of researches report the increase in antimicrobial activity with the decrease in degree of polymerization (i.e. molecular weight) [3]. It is expected that adhesion of HMW antimicrobial polymers to negatively charged bacterial cell membrane should be highly effective than adhesion of LMW antimicrobial polymers. Instead, contradicting results are found according to which LMW antimicrobial polymers have greater biocidial activity [4].
3. In the last part of the manuscript, the author described a new method which might help to overcome AMP delivery issues. Is this the only effective method? Could authors described more about it?
REPLY: Other than uncontrolled release rate, factors such as low biocompatibility, low metabolic stability, low solubility, high susceptibility to degradation and toxicity act as a hurdle in applications of antimicrobial peptides [215]. Incorporation of antimicrobial peptides into nanostructure can work as solution regarding these issues. Some of the nanosystems re-searched are metal nanoparticles [216], carbon nanotubes [217], liposomes [218], liquid crystalline particles [219], dendritic system [220], aptamers [221], hydrogels [222], poly-mers [223] and cyclodextrin [224].
Reviewer 2 Report
The authors reviewed many aspects of antimicrobial peptides (AMPs) including the type, the mode of action, factors impacting activities, origin of AMPs, and some biomedical applications. This review manuscript covered many aspects of AMPs that could potentially be beneficial for researchers working in this field.
However, this manuscript has several major issues and inappropriate discussion. Several sections are not fully supported by the reference, or the discussion is not aligned with the section title/purpose. A couple of misleading discussion can be problematic for this manuscript to be published. I would not recommend accepting this manuscript until all the discussion is properly developed and supported by adequate reference.
Line 28-29: The author started the introduction by claiming ambient condition is favorable for most of the living organisms including microbials. The authors must be specific on the term “ambient”. If it is only ambient temperature and the relative humidity or pH is extreme, the starting claim is not valid at all.
Line 33-34: sentence needs grammar check
Line 42: authors should define AMP as this is the first appearance in the main manuscript
Line 53-56: the authors tried to claim that alpha helical structure of AMP is all due to the presence of intramolecular disulfide bound. This is a very misleading claim. Many AMPs and cell penetrating peptides such as Cecropin B, MAP, Pep-1, Tp-10 form the helix because of the amphiphilic nature, not due to disulfide bond. Please read the following reference and revise this claim.
Secondary structure of cell-penetrating peptides during interaction with fungal cells. Protein Science, 27: 702-713
Line 81-86: The authors tried to claimed that the example from reference #35 was to support that the secondary structure can impact the interaction between AMPs and cells. However, the substitution to Lys and Arg is related to charge, and change Try is related to hydrophobicity. The author did not correlate these modifications to and structural change. This whole section is not a valid stand-alone section, rather should be integrated into other specific sections.
Line 96-104: this section is very misleading. If the intent is to explain how hydrophobicity can impact the AMP interaction with target cells, the authors should use reference that has examples that AMP and its variants with different hydrophobicity has different antimicrobial activities, in a systematic discussion. Authors need to find my reference for this section. Consider this example:
Copolovici DM, Langel K, Eriste E, Langel U. Cell-penetrating peptides: design, synthesis, and applications. ACS Nano.
Karagiannis ED, Urbanska AM, Sahay G, et al. Rational design of a biomimetic cell penetrating peptide library. ACS Nano. 2013;7(10):8616-8626.
Line 167, Table 2: the authors listed several AMPs that has antifungal activities. The list of AMPs is very limited. Consider this is a review article, authors should provide more examples of AMPs that work on fungal cells, preferably via different mode of action (MoA). See reference for example:
Translocation of cell-penetrating peptides into Candida fungal pathogens. Protein Science, 26: 1714-1725
Assessing the uptake kinetics and internalization mechanisms of cell-penetrating peptides using a quenched fluorescence assay. Biochim Biophys Acta Biomembr. 2010;1798(3):338-343.
Comparison of the interaction, positioning, structure induction and membrane perturbation of cell-penetrating peptides and non-translocating variants with phospholipid vesicles. Biophys Chem. 2003;103(3):271-288.
Line 180-190: the authors listed AMPs that can target cancer cell(lines). The word “treatment” is very misleading. If the examples that authors provided are actually in clinical trial, it should be considered as investigational drug. If the AMPs are only tested in vitro with cell lines, not in vivo with animal models or even in human trial, it should never be called “treatment”. This section is very misleading. AMPs may only have cytotoxicity to cancer cell lines, but it is too far away to call it a cancer treatment.
Line 305: typo, C. albicans
Multiple locations, S. aureus, K. pneumoniae, C. albicans need to be italic
Author Response
Thank you for your valuable suggestions I have made changes as per your suggestions.
1. Line 28-29: The author started the introduction by claiming ambient condition is favorable for most of the living organisms including microbials. The authors must be specific on the term “ambient”. If it is only ambient temperature and the relative humidity or pH is extreme, the starting claim is not valid at all.
REPLY: This statement is removed from the manuscript.
2. Line 33-34: sentence needs grammar check
REPLY: Statement is modified to- And the microorganisms having harmful effect cause infections or diseases such as anthrax, conjunctivitis, ring worm, influenza and many more
3. Line 42: authors should define AMP as this is the first appearance in the main manuscript
REPLY: I have considered this by moving AMP introduction paragraph before table.
4. Line 53-56: the authors tried to claim that alpha helical structure of AMP is all due to the presence of intramolecular disulfide bound. This is a very misleading claim. Many AMPs and cell penetrating peptides such as Cecropin B, MAP, Pep-1, Tp-10 form the helix because of the amphiphilic nature, not due to disulfide bond. Please read the following reference and revise this claim.
REPLY: Α α helical AMPs are those which are disorderly aligned under aqueous surrounding and form helical structure during lipid interaction (example: LL-37) [14].
5. Line 81-86: The authors tried to claimed that the example from reference #35 was to support that the secondary structure can impact the interaction between AMPs and cells. However, the substitution to Lys and Arg is related to charge, and change Try is related to hydrophobicity. The author did not correlate these modifications to and structural change. This whole section is not a valid stand-alone section, rather should be integrated into other specific sections.
REPLY: Secondary structure plays a significant role in improving the antimicrobial activity of antimicrobial peptides. It is observed that among all the structures, α helical structure promotes the insertion of antimicrobial peptides into the cellular membrane due to its facially amphiphillic structure [40]. In an investigation two antimicrobial peptides were compared, C5A (single helix) and AH (structure consist of a hinge in between two short helix) among which C5A had more uniform helical structure. Analysis showed that 10 nm concentration of C5A was capable of lysing lipid vesicles whereas, AH showed comparatively less potency [41].
6. Line 96-104: this section is very misleading. If the intent is to explain how hydrophobicity can impact the AMP interaction with target cells, the authors should use reference that has examples that AMP and its variants with different hydrophobicity has different antimicrobial activities, in a systematic discussion. Authors need to find my reference for this section. Consider this example:
REPLY: Microbial membranes are protected from the exogenous matters such as polysaccharides, proteins and peptides due its hydrophobic characteristics, but AMP has the ability to in-teract with microbial membrane [49]. In an investigation, stearyl, lauryl, cholyl and choleosytryl peptides were formulated by addition of four hydrophobic moieties which are stearic acid, lauric acid, cholic acid and cholesteryl chloroformate to arginine rich peptide respectively. Among these peptide, stearly peptide showed the highest transfec-tion efficiency which was reported as an effect of high hydrophobicity of stearyl peptide [50].
7. Line 167, Table 2: the authors listed several AMPs that has antifungal activities. The list of AMPs is very limited. Consider this is a review article, authors should provide more examples of AMPs that work on fungal cells, preferably via different mode of action (MoA). See reference for example:
REPLY: Three AMPs are added: Penetratin, APP and polybia-CP.
8. Line 180-190: the authors listed AMPs that can target cancer cell(lines). The word “treatment” is very misleading. If the examples that authors provided are actually in clinical trial, it should be considered as investigational drug. If the AMPs are only tested in vitro with cell lines, not in vivo with animal models or even in human trial, it should never be called “treatment”. This section is very misleading. AMPs may only have cytotoxicity to cancer cell lines, but it is too far away to call it a cancer treatment.
REPLY: Table 5: List of some of the antimicrobial peptides that can be used in cancer therapeutic study.
9. Line 305: typo, C. albicans
REPLY: Corrected
10. Multiple locations, S. aureus, K. pneumoniae, C. albicans need to be italic
REPLY: Corrected
Reviewer 3 Report
The submitted manuscript “Antimicrobial Peptides and Their Applications in Biomedical Sector” by Sultana et al highlighted all the major functions and applications of antimicrobial peptides (AMPs) in the biomedical field and concludes the future potential of AMPs. It is a very interesting summary. However, there are multiple excellent reviews on AMPs published online (Lancet Infect Dis 2020; 20: e216–30, https://doi.org/10.1016/S1473-3099(20)30327-3, Chem. Soc. Rev., 2021,50, 4932-4973 https://doi.org/10.1039/D0CS01026J), how can they differentiate their submitted review to the current ones. At this stage, I recommend this manuscript be published after major revision.
There are few minor comments,
- Some repeating expressions or abbreviations, for example, antimicrobial peptides (AMPs) was frequently used in the text but not consistent.
- Page 3 line 100, there is a lack of reference to support the statement “Two AMPs La47 and Css54 have chemically synthesized arachnid venoms and their antimicrobial activity was analysed in the presence of antibiotics”.
- Page 4 table 1, it is better to include the antibacterial activity inside the table.
- The authors should carefully cite the proper references for their AMPs, such as Onc112 in table 1, it is developed by Hoffmann et al (Journal of Medicinal Chemistry 2010 53 (14), 5240-5247 DOI: 10.1021/jm100378b, and Angew. Chem. Int. Ed., 53: 12236-12239. https://doi.org/10.1002/anie.201407145)
- page 6 table 3, the peptide they listed, HaA4 is not antiviral. The authors should double-check all the peptides they referred to and listed in this manuscript.
Author Response
Thank you for your valuable suggestions I have made changes as per your suggestions.
1. Some repeating expressions or abbreviations, for example, antimicrobial peptides (AMPs) was frequently used in the text but not consistent.
REPLY: Corrected
2. Page 3 line 100, there is a lack of reference to support the statement “Two AMPs La47 and Css54 have chemically synthesized arachnid venoms and their antimicrobial activity was analysed in the presence of antibiotics”.
REPLY: This statement has been removed
3. Page 4 table 1, it is better to include the antibacterial activity inside the table.
REPLY: This column has been incorporated
4. The authors should carefully cite the proper references for their AMPs, such as Onc112 in table 1, it is developed by Hoffmann et al (Journal of Medicinal Chemistry 2010 53 (14), 5240-5247 DOI: 10.1021/jm100378b, and Angew. Chem. Int. Ed., 53: 12236-12239. https://doi.org/10.1002/anie.201407145)
REPLY: Reference has been corrected
5. page 6 table 3, the peptide they listed, HaA4 is not antiviral. The authors should double-check all the peptides they referred to and listed in this manuscript.
REPLY: This peptide has been removed
Round 2
Reviewer 2 Report
Line 53: first time introduce antimicrobial peptides, should define the abbreviation here, as AMP showed in Line 63 without previous definition.
Line 64: the authors claimed alpha helix is disorderly aligned structure. It is not an appropriate statement. The helical structure should be viewed as an ordered structure. Also, AMP may form the helical structure in the aqueous phase, prior to membrane engagement.
Line 140-146: it is a great example to show that modified arginine-rich peptides have different translocation efficacy. However, if the intent is to review how the hydrophobicity of the AMP itself can impact the translocation, reference focusing on the amino acid impact should be considered.
Line 228-237: similar to the original comment regarding the usage of the word "treatment". It is still not appropriate to claim AMP is a "treatment" to cancer. It should be only considered as AMP has anti-cancer cell activity. Meanwhile, authors should listed reference to show while AMP can target cancer cells, it would not damage the normal tissue cells. In vitro or in vivo results should work.
Author Response
Line 53: first time introduce antimicrobial peptides, should define the abbreviation here, as AMP showed in Line 63 without previous definition.
Reply: This definition of AMP I have considered for first time introduction: Antimicrobial peptides are short chain (5 to 100) of amino acids which possess the responsibility to counter microbial attack or any infective agent in all the living organisms [7].
Line 64: the authors claimed alpha helix is disorderly aligned structure. It is not an appropriate statement. The helical structure should be viewed as an ordered structure. Also, AMP may form the helical structure in the aqueous phase, prior to membrane engagement.
Reply: Corrected- Α α helical antimicrobial peptides possess about 0.15 nm distance between two adjacent bond angle (example: LL-37).
Line 140-146: it is a great example to show that modified arginine-rich peptides have different translocation efficacy. However, if the intent is to review how the hydrophobicity of the AMP itself can impact the translocation, reference focusing on the amino acid impact should be considered.
Reply: For instance, magainin has the ability to inhibit the growth of gram negative bacteria but the analogs with improved hydrophobicity have shown effectiveness against gram positive bacteria as well [50].
Line 228-237: similar to the original comment regarding the usage of the word "treatment". It is still not appropriate to claim AMP is a "treatment" to cancer. It should be only considered as AMP has anti-cancer cell activity. Meanwhile, authors should listed reference to show while AMP can target cancer cells, it would not damage the normal tissue cells. In vitro or in vivo results should work.
Reply: Anticancer activity is replaced by word inhibits cancer cell growth
Reviewer 3 Report
The authors have only addressed the minor comments. However, the major concerns have not been addressed: "There are multiple excellent reviews on AMPs published online (Lancet Infect Dis 2020; 20: e216–30, https://doi.org/10.1016/S1473-3099(20)30327-3, Chem. Soc. Rev., 2021,50, 4932-4973 https://doi.org/10.1039/D0CS01026J), how can they differentiate their submitted review to the current ones?
Author Response
The authors have only addressed the minor comments. However, the major concerns have not been addressed: "There are multiple excellent reviews on AMPs published online (Lancet Infect Dis 2020; 20: e216–30, https://doi.org/10.1016/S1473-3099(20)30327-3, Chem. Soc. Rev., 2021,50, 4932-4973 https://doi.org/10.1039/D0CS01026J), how can they differentiate their submitted review to the current ones?
Yes, I agree there are various available paper on the similar topic including the one which is mentioned in the comment. This paper is different because it includes the implementation techniques beside the introduction and important functions of antimicrobial peptides. This paper is useful for those who are studying the importance of AMP and methods to implement in biomedical applications and also it consist of discussion about smart and intelligent delivery.
Round 3
Reviewer 2 Report
Thank you for addressing the comment. I would recommend this manuscript for publication.
Author Response
Thankyou for your valuable suggestions which has helped me to improve my manuscript.
Reviewer 3 Report
The authors responded to the comment: "There are multiple excellent reviews on AMPs published online (Lancet Infect Dis 2020; 20: e216–30, https://doi.org/10.1016/S1473-3099(20)30327-3, Chem. Soc. Rev., 2021,50, 4932-4973 https://doi.org/10.1039/D0CS01026J), how can they differentiate their submitted review to the current ones?" However, for the sake of the reader of a wider community, it will be beneficial to add the response into the main text of the manuscript.
Author Response
This article is different from various other papers on antimicrobial peptides because it not only includes information about the factors affecting the functioning of antimicrobial peptides and its important role but along with it, focuses on the implementation techniques of antimicrobial peptide in biomedical sector. In this review, smart and intelligent delivery methods are also considered. Thus, this review could be helpful for readers which require compact information of antimicrobial peptide’s role in biomedical applications.